# Solar cell designs by maximizing energy production based on machine learning clustering of spectral variations

J. M. Ripalda [1], J. Buencuerpo [1,3] & I. García [2]

Due to spectral sensitivity effects, using a single standard spectrum leads to a large uncertainty when estimating the yearly averaged photovoltaic efficiency or energy yield. Here we demonstrate how machine learning techniques can reduce the yearly spectral sets by three orders of magnitude to sets of a few characteristic spectra, and use the resulting proxy spectra to find the optimal solar cell designs maximizing the yearly energy production. When using standard conditions, our calculated efficiency limits show good agreement with current photovoltaic efficiency records, but solar cells designed for record efficiency under the current standard spectra are not optimal for maximizing the yearly energy yield. Our results show that more than $1\,\mathrm{MWh\,m^{-2}\,year^{-1}}$ can realistically be obtained from advanced multijunction systems making use of the direct, diffuse, and back-side albedo components of the irradiance.

[1] Instituto de Micro y Nanotecnología, IMN-CNM, CSIC (CEI UAM+CSIC) Isaac Newton, 8, 28760 Tres Cantos, Madrid, Spain. [2] Instituto de Energía Solar, Universidad Politécnica de Madrid, Avda. Complutense 30, 28040 Madrid, Spain. [3]Present address: National Renewable Energy Laboratory, Golden, CO 80401, USA. Correspondence and requests for materials should be addressed to J.M.R. (email: j.ripalda@csic.es)

The expansion of photovoltaics (PV) used to be constrained by the high cost of solar cells, but the cost of PV electricity is now mostly determined by area-related costs other than the cost of the solar cells[1], thus increasing the energy efficiency not only results in a higher return on investment, but also lessens the environmental and esthetic impact of PV installations. The nominal or standard efficiency of a solar cell is defined as the electrical power output per unit area in standard test conditions divided by the standard value of the global horizontal irradiance (or direct normal irradiance in the case of concentrator solar cells). In practice, the standard efficiency differs from the yearly averaged efficiency, as determined by the yearly energy yield per unit area divided by the time integrated solar irradiance, due to spectral variations as a function of the position of the sun and atmospheric phenomena. The nominal standard efficiency of modules in utility-scale new installations is increasing by 0.6% per year on an average[1]. At the current rate we will reach the practical limits of single junction photovoltaic technology within a decade. A similar trend towards higher inverter efficiencies has also been reported, and 80% of the U.S. utility-scale systems installed in 2016 used tracking[1]. Because the capacity of a PV installation is the product of many factors (cell efficiency, inverter efficiency, tracking, optical efficiency) the trends towards higher efficiency in each of these factors reinforces the others in a synergistic nonlinear positive feedback loop. Silicon wafer costs currently represent 8.6% of utility system costs and an even smaller fraction of the levelized cost of energy[1]. As a consequence, there is almost no margin left to compete in terms of solar cell costs, and demand for emerging PV technologies can only be expected if these can exceed the efficiency of silicon single junctions. The only proven method to significantly increase the efficiency beyond the limits of conventional silicon technology is the use of multijunction devices, used either with or without optical concentration, but there still exists uncertainty about how the changes of the solar spectrum as a function of time affect the energy production of multijunction solar cells[2–4].

Here we demonstrate that data sets with thousands of solar spectra can be reduced to a few characteristic proxy spectra using machine learning techniques, and successfully use these proxy spectra to predict the yearly averaged efficiency as a function of the solar cell design.

## Results

**Binning and clustering.** A method to estimate the yearly energy yield was proposed by García et al., where spectra are grouped or binned according to their spectral characteristics and then all spectra in the same group or bin are averaged to obtain a few representative spectra[2]. The binning method is applied here and compared with a clustering technique where spectra are classified on the basis of their Euclidean distances in a highly multi-dimensional vector space defined by the number of components of the spectra. We use a machine learning technique known as structured feature agglomerative clustering[5], where the characteristic features of the spectra (such as absorption and transmission bands) are identified by searching for correlations as a function of time. This is used as a dimensionality reduction step for computational efficiency, but the core of our method is the widely used k-means clustering algorithm. A visualization of the clustering method is shown in Fig. 1. The machine learning method, the detailed balance solar cell model, and the spectral data set are described in detail in the Methods section. Other machine learning techniques are discussed in Supplementary Note 1.

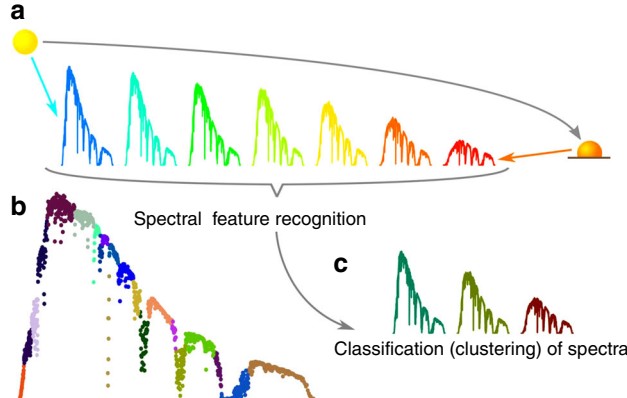

**Fig. 1** Visualization of the method. **a** The solar spectra depend on the position of the sun and atmospheric phenomena. We reduce yearly spectral sets with $2 \times 10^4$ solar spectra to a few characteristic proxy spectra. **b** The main features of the spectra are identified using structured feature agglomerative clustering as an unsupervised dimensionality reduction step. A set of 20 automatically identified features is shown in this example. **c** Spectra with the most similar features are then clustered together into a small set of proxy spectra using the k-means method

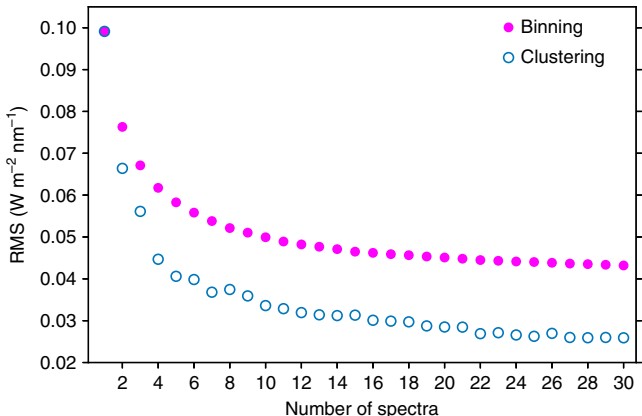

**Fig. 2** Spectral classification error. Root mean square statistics relative to the cluster (or bin) center for the clustering and binning methods as a function of the number of clusters (or bins)

**Validation and convergence.** Both the binning and the clustering method reduce the number of spectra by averaging similar spectra, but differ on the criteria used to merge spectra. A useful measure of the classification error is the root mean square statistic relative to the cluster or bin center shown in Fig. 2. While the binning method is deterministic, always producing the same result for a given data set, the machine learning method is stochastic, as revealed by the random fluctuations that appear to be a function of the number of clusters, but in fact are determined by the initial random seed of the method.

To validate the proposed technique, we calculate the yearly average efficiency for a number of random but nearly optimal (within 2% of the efficiency maximum) six-junction band gap combinations. In Fig. 3 we plot the difference between the results obtained with a reduced set of proxy spectra and the result obtained using the full yearly data set with $2 \times 10^4$ spectra. The results obtained using the binning technique (Fig. 3a), which had previously only been tested with a single four-junction device design[2], are compared with results from the clustering technique (Fig. 3b). When only a small set of proxy spectra can be used, both methods are found to yield an efficiency overestimate of the

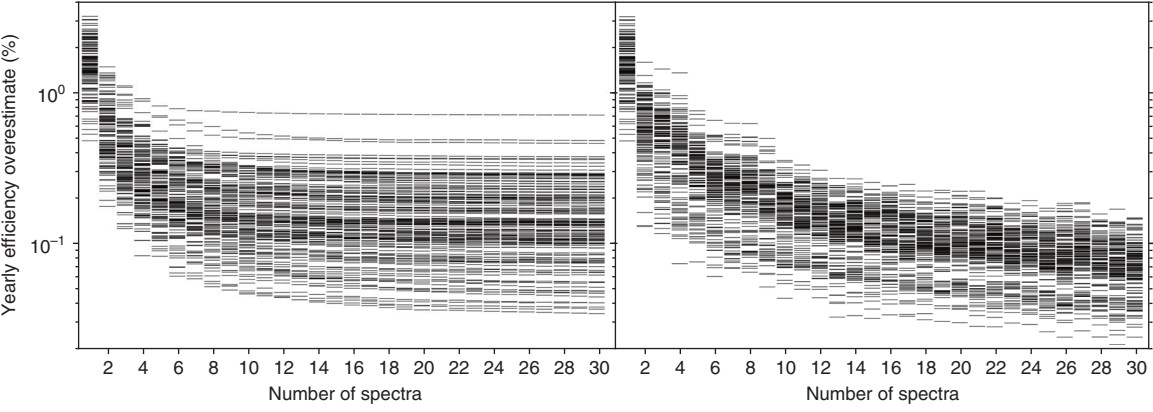

**Fig. 3** Validation and convergence. Scatter plots of the yearly averaged efficiency overestimate as a function of the number of proxy spectra. The data correspond to a set of random, but near optimal (within 2% of the maximum efficiency), series connecting six-junction devices. **a** Binning method. **b** Clustering method

same magnitude, but the clustering method is preferable when the number of proxy spectra is larger than 6, as smaller overestimates (<0.3%) are obtained using the clustering technique (Fig. 3b).

**Energy yield as a function of the band gaps**. As reviewed by Kurtz et al., there have been many previous efforts to optimize solar cell band gaps, but in all cases the target of the optimization was aimed at maximizing the efficiency under standard conditions[6,7]. To illustrate the possible uses of our machine learning results, we have optimized the band gaps by maximizing the energy yield, instead of the standard efficiency. This is a more economically relevant target, which in principle can only be achieved by using computationally intensive resources and methods. The approach here discussed provides a way to overcome these limitations while still keeping high precision in the results. As will later be seen, the band gaps that optimize the energy yield differ significantly from those that optimize the standard efficiency.

In the following results we use 15 clusters resulting from a data set of synthetic spectra obtained with the SMARTS 2.9.5 code at a fixed latitude of 40°N[8], and the detailed balance theory including radiative coupling, photon recycling, and nonradiative recombination to compute the yearly energy yield of multijunction devices[9–13] (see Methods section for further details). Results for flat plate devices at 1 sun are shown in Fig. 4. As illustrated in Fig. 4b with two examples, points with the same efficiency value and the same color (average photocurrent) are junctions belonging to the same tandem device. These plots can be used to find out how to adjust the band gaps to minimize the efficiency loss when the optimal band gap combination is not attainable, and to quantify such efficiency loss. Several local minima can be found within 2% of the global maximum. The band gap of the bottom junction largely determines the short circuit current and the optimal band gaps for the other junctions. Bottom junctions placed at the low energy edge of the A, B, C and D atmospheric transmission bands result in different local efficiency maxima. Figure 4c shows three junction multiple terminal devices with a mechanically stacked and electrically isolated silicon bottom junction, as experimentally demonstrated by Essig et al.[14]. In this case the current represented by the color scale is the sum of the currents in both mechanically stacked devices. As a consequence, high band gap combinations have a higher current in the silicon device and a higher total current. Devices with multiple terminals are much less sensitive to the band gap energies than current

matched devices, allowing for greater flexibility in the choice of band gaps.

Figure 5 shows the dependence of the energy yield on the band gaps for series connected devices at 1000 suns. As the number of junctions is increased, the band gaps of the top junctions shift towards higher energies, while gaps of the bottom junctions remain pinned at the low energy edges of the infrared transmission bands. Due to practical constraints, top junction band gap energies lower than those here reported as optimal might have in practice a higher energy yield[15]. The energy yield of the optimal six-junction series connected device is 911.6 kWh m$^{-2}$ year$^{-1}$ using synthetic spectra obtained with the SMARTS 2.9.5 code. If measured spectra from the National Solar Resource DataBase are used, the same device yields 1139.9, 897.6, 739.3 and 744.7 kWh m$^{-2}$ year$^{-1}$, at Reno, Boulder, Indianapolis, and Philadelphia, respectively[16]. All these locations are at the same latitude used to generate the synthetic spectra (40°N); thus, the observed differences are mostly attributed to differences in cloud cover and other atmospheric phenomena, but also to differences in height above sea level and ambient temperatures.

Table 1 shows the optimal band gap combinations in order of increasing yearly energy yield, assuming dual axis tracking in all cases. The yearly averaged efficiency (Ef.) is compared with the standard efficiency (Std. Ef.) and current efficiency records (Rec.)[14,17,18]. Some of the entries correspond to low current, high band gap (HG) local efficiency maxima. For multiple terminal, mechanically stacked devices, the number of junctions is expressed as the number of junctions in the top device + number of junctions in the bottom device. Gaps for the experimental efficiency records are near optimal in most cases, but do not correspond exactly to the here reported optimal gaps. The contribution of separate bifacial silicon devices (Bi) at the back of the concentrator modules is only considered in the energy yield calculations, not in the efficiency values. Our calculated efficiencies under standard conditions (25 °C operation temperature and ASTM G173 spectra) are close to the corresponding efficiency records (except for the recent silicon-based triple junction by Cariou et al., with 33.3% efficiency[18]). The effects of spectral variability cause the yearly averaged efficiency to be lower than the standard efficiency. In contrast with previously published optimal band gaps[6,19], our results show significantly higher bottom junction band gaps (no optimal bottom junction is found to lie on the A transmission band, 0.50–0.69 eV) and lower energy top junction band gaps. The red-shifts are approximately 30 meV for the top junction, while middle cell band gaps show red-shifts of the order of 10–20 meV relative to the gaps obtained

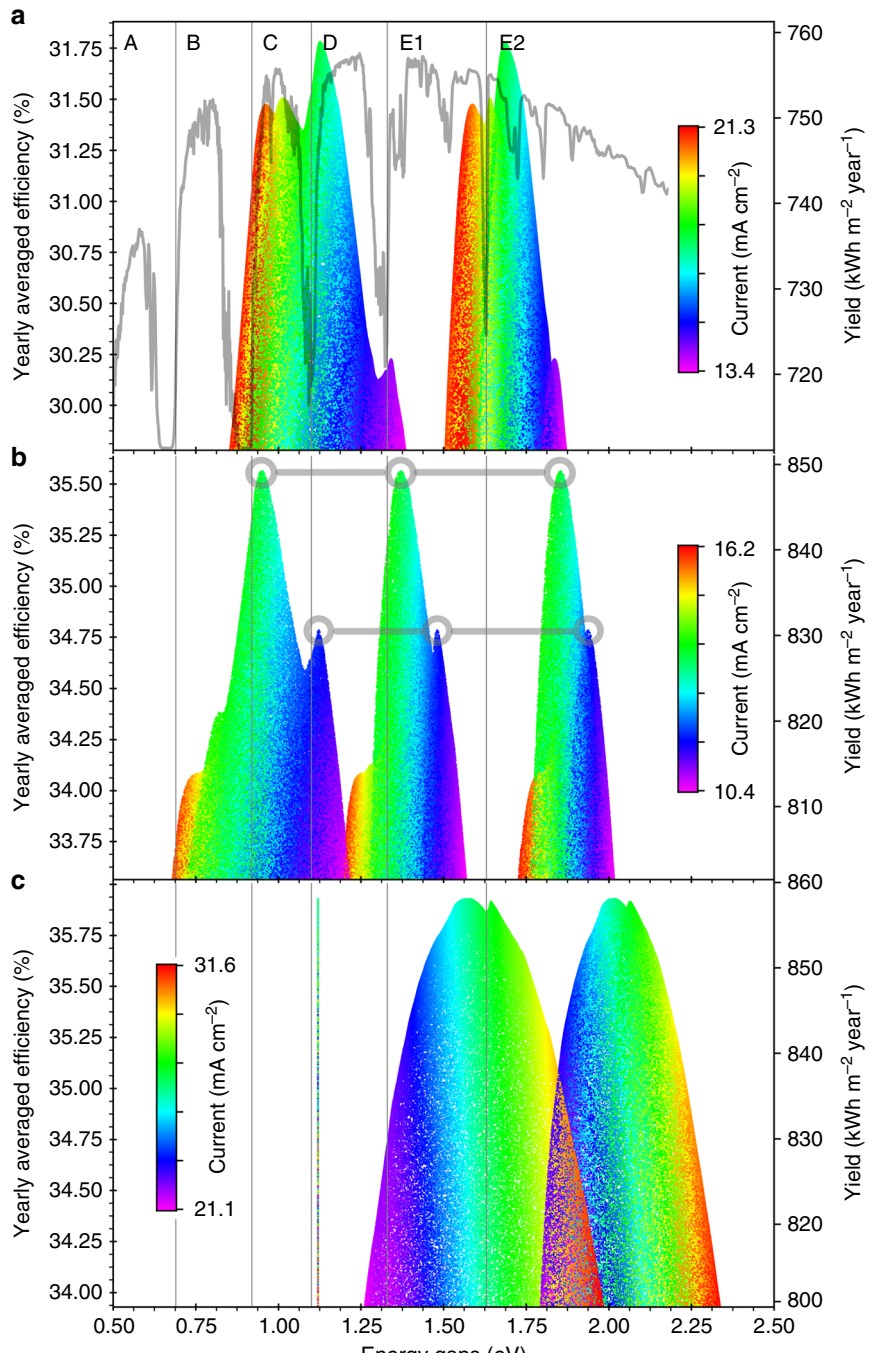

**Fig. 4** Energy yields at 1 sun. Labels on top correspond to the low energy thresholds of each of the atmospheric transmission bands. **a** Two junctions. The standard global spectral irradiance is overlaid to highlight the alignment of the optimal band gaps with the transmission bands in the spectrum. **b** Three junctions. Two sets of near optimal band gaps are highlighted to illustrate that points with the same efficiency value and the same color (average photocurrent) are junctions belonging in the same tandem device. **c** Three junctions with a mechanically stacked and electrically isolated monofacial silicon bottom junction

using the standard spectrum. A similar, but larger, red-shift was reported by García et al. when using an experimental yearly spectral data set based on observations at Golden CO, a location that is expected to be a close match to the ASTM G173 standard[2]. Solar cells designed for record efficiency under the ASTM G173 standard spectra are therefore not optimal for maximum yearly energy yield, and there might be good reason to define a new standard based on at least two proxy spectra so that new record solar cell developments are optimal for maximum yearly energy yield. Another advantage of defining a new standard

would be related to the fact that energy band gaps above 2 eV can currently only be reached at the expense of material quality, compromising quantum efficiency and voltage. This problem is alleviated by the fact that the top junction band gaps that maximize the yearly energy yield are red-shifted relative to those that maximize the efficiency under the current standard.

Extending the spectral response to long wavelengths implies a number of trade-offs such as the need to increase the lattice parameter of the bottom junctions, reduced material quality, and increased difficulty in the design of antireflective coatings. But

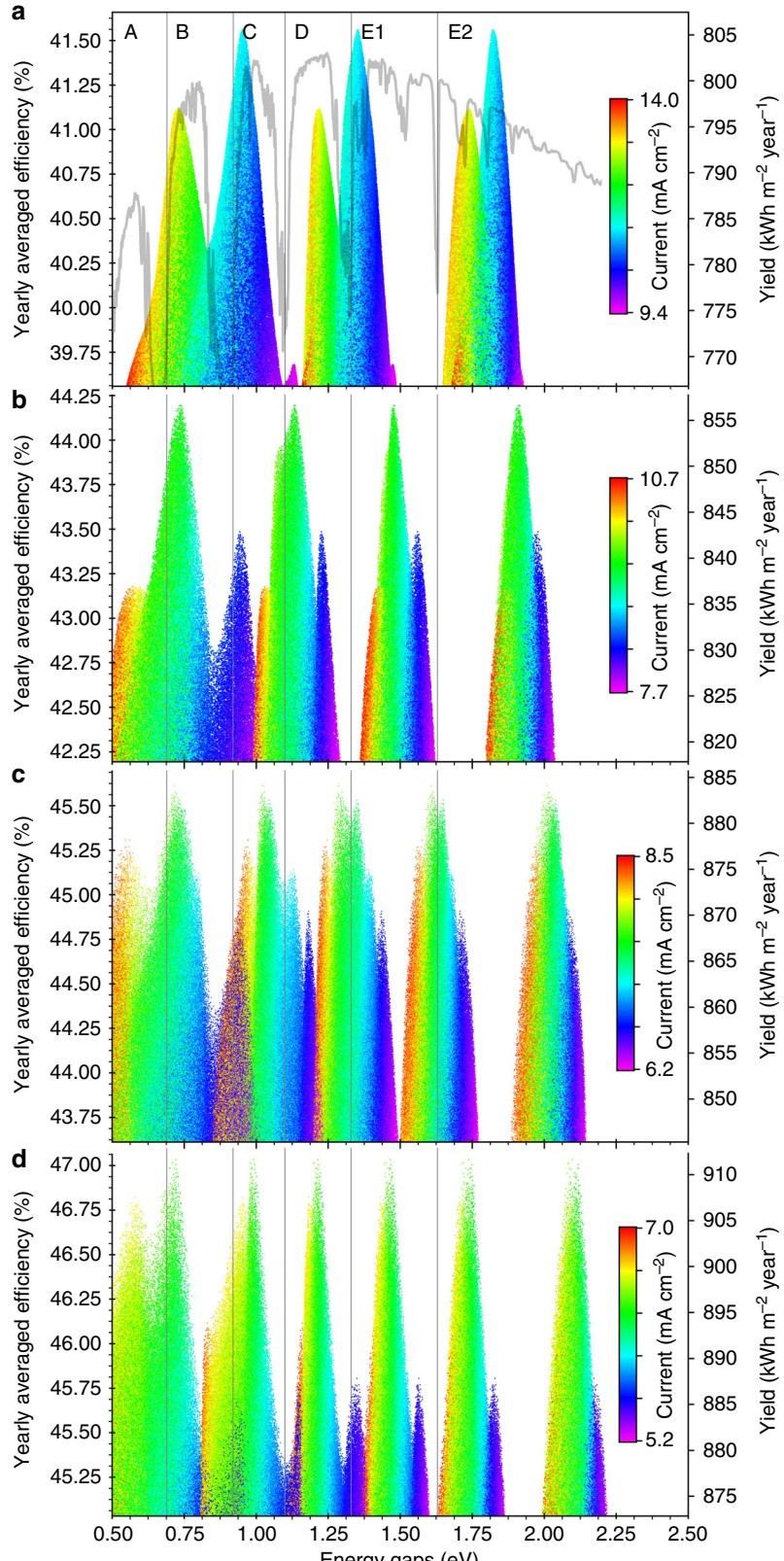

**Fig. 5** Energy yields at 1000 suns. **a** Three junctions. The standard direct spectral irradiance is overlaid to highlight the alignment of the optimal band gaps with the transmission bands in the spectrum. **b** Four junctions. **c** Five junctions. **d** Six junctions

**Table 1 Solar cell designs in order of increasing yearly energy yield**

| Junctions | Suns | kWh m$^{-2}$ year$^{-1}$ | Ef. (%) | Std. Ef. (%) | Rec. (%) | Gaps (eV) |
|---|---|---|---|---|---|---|
| 1 Diffuse | 1 | **106.3** | 27.76 | — | — | 1.42 Diffuse light only |
| 1 Si | 1 | **617.7** | 25.87 | 26.98 | 26.7 | 1.12 Gap not optimal, EQE = 1 |
| 1 | 1 | **648.2** | 27.14 | 27.84 | 28.8 | 1.35 ERE = 0.2 |
| 2 | 1 | **758.0** | 31.81 | 33.39 | 32.8 | 1.13, 1.69 |
| 1 + 1 | 1 | **792.6** | 33.26 | 34.32 | 32.8 | 1.13, 1.81 |
| 3 | 1000 | **805.5** | 41.56 | 44.31 | 44.4 | 0.95, 1.35, 1.82 |
| 3 HG | 1 | **830.6** | 34.78 | 36.88 | 33.3 | 1.12, 1.48, 1.94 |
| 3 | 1 | **849.3** | 35.63 | 37.65 | 37.9 | 0.95, 1.38, 1.86 |
| 4 | 1000 | **856.6** | 44.19 | 47.15 | 46.0 | 0.74, 1.14, 1.48, 1.91 |
| 5 HG | 1000 | **870.4** | 44.90 | 47.71 | — | 0.94, 1.18, 1.44, 1.71, 2.09 |
| 2 + 2 | 1000 | **879.6** | 45.38 | 47.56 | — | 0.73, 1.14, 1.54, 1.97 |
| 5 | 1000 | **884.1** | 45.61 | 49.11 | — | 0.72, 1.02, 1.29, 1.61, 2.01 |
| 6 | 1000 | **911.6** | 47.03 | 50.19 | — | 0.70, 0.98, 1.21, 1.46, 1.74, 2.11 |
| 3 + 3 | 1000 | **939.8** | 48.48 | 51.21 | — | 0.71, 1.00, 1.25, 1.56, 1.83, 2.19 |
| Bi + 2 | 1 | **1007.9** | 36.00 | 37.62 | 35.9 | 1.12 + 1.58, 2.02 |
| Bi + 5 HG | 1000 | **1116.8** | 44.90 | 47.71 | — | 1.12 + 0.94, 1.18, 1.44, 1.71, 2.09 |

Mechanical stacking is denoted by a plus sign

Ef. is the yearly averaged efficiency, Rec. is the experimental record efficiency[17], HG is the high band gap local efficiency maximum, Bi is the energy yield of concentrator modules including bifacial silicon solar cells in the back of the module for collecting diffuse and albedo irradiance[20]

tandems based on high band gap bottom junctions (HG) have nearly the same efficiency with lower short circuit currents. This could be advantageous for very high concentration devices, as resistance losses scale as the current squared.

As shown in Table 1, the use of concentration does not necessarily lead to a higher energy yield. The advantage of optical concentration is mostly one of sustainability and cost when using scarce materials and advanced cell technologies such as III–V semiconductors.

The high energy yield for silicon devices is due to the assumption of dual axis tracking for all calculations in Table 1. Without tracking the calculated yearly energy yield for silicon devices is 469.0 kWh m$^{-2}$ and 582.2 kWh m$^{-2}$ with single axis azimuthal tracking. Even though the gap of silicon is not optimal, this is compensated by the fact that the external quantum efficiency (EQE) of state of the art silicon devices is approximately 1 in the whole spectral range.

Benitez et al. have proposed multijunction concentrator modules with bifacial silicon solar cells covering the whole back plane of the module[20]. The energy yield increase due to back-side albedo irradiance in bifacial cells can reach 50%, but depends on a number of factors[21]. For two axis tracking, low ground cover ratios, panel height larger than the panel dimensions, and a 0.8 albedo, the energy yield increase due to bifacial operation is 25%[22]. This factor has been multiplied by the energy yield of monofacial flat plate silicon in Table 1 to obtain an optimistic upper limit to the energy yield increase due to back-side albedo irradiance. As shown in Table 1, the combination of bifacial and concentrator technology leads to the highest photovoltaic energy yields practically attainable due to the additional back-side albedo contribution of 154.4 kWh m$^{-2}$ per year, and the diffuse sky irradiance contribution of 92 kWh m$^{-2}$ per year.

## Discussion

There is a need for solar cell testing standards that better reflect the yearly averaged efficiency, as solar cells designed for record efficiency under the ASTM G173 standard spectra are not optimal for maximum yearly energy yield. The methods here discussed, or a variation thereof, will enable accurate determinations of the yearly energy yield using a few characteristic spectra. Using these methods we show that technologies integrating multijunctions with silicon bifacial single junctions are capable of energy yields of more than 1 MWh m$^{-2}$ per year by harvesting the direct, diffuse, and back-side albedo components of the irradiance.

## Methods

**Spectral data set**. All parameters that are not explicitly mentioned here are those corresponding to the ASTM G173 standard[23]. We have obtained data sets of synthetic spectra with $2 \times 10^4$ spectra per year from the SMARTS 2.9.5 code at a fixed latitude of 40°N[8], as this latitude is a good match to the current standard. The distance between trackers has an effect on the yearly energy yield due to shadowing[24]. To consider this effect we have used the same approach used by Villa and Marti, where the maximum usable air mass (AM) is limited to 4 (sun near the horizon), corresponding to a maximum solar zenith angle of 75.5°[19]. This is roughly equivalent to a ground cover ratio of 13% assuming a total tracker shutdown in the case of partial shadowing, or a correspondingly higher ground cover ratio in case of a higher shadow tolerance[24].

Jaus and Gueymard have published statistics on 6.9 million measurements of aerosol optical depth (AOD) and precipitable water (PW) from 379 sites of the worldwide AERONET network[25]. To include the effects of variability of the atmospheric conditions in our simulations, the AOD and PW histograms presented by Jaus and Gueymard are here fitted by a lognormal and chi squared distribution respectively. Random AOD and PW values have been sampled from said model distributions when generating random spectra.

Diffuse spectra have been obtained by subtracting the direct normal irradiance from the global normal irradiance collected by a sun tracking surface. The subtraction has been done prior to clustering to preserve the temporal correlation between global and direct spectra.

The seasonal variations of the clouds have an effect on the statistics of the spectra. The typically cloudy skies in winter will block a higher proportion of winter spectra, and these have a higher air mass than summer spectra. This effect is automatically included when using experimental spectra, but not when using synthetic spectra. After comparing with results obtained using experimental spectra from Boulder and other locations with 40°N latitude from the National Solar Resource DataBase[16], we have reduced the energy yield obtained with synthetic spectra by corrective factors to account for the energy loss caused by clouds and other effects not included in the synthetic spectra. These factors are 0.74, 0.75, and 0.80 for devices harvesting direct, global, and diffuse irradiance, respectively. The choice of Boulder as a reference is somewhat conservative, as the cloud cover is much lower in areas where concentrators are of most interest (e.g.: Arizona, Nevada, and Australia).

**Machine learning and binning**. In the clustering method the dimensionality of the vector space has been reduced using structured feature agglomerative clustering with a connectivity constraint matrix to ensure that only contiguous points are included in each spectral feature. We have found that 24 spectral features suffice to give a complete description of the whole data set, but have chosen to use 200 spectral features to ensure that no significant information is lost in this step. The resulting spectral features correspond to the Fraunhofer absorption bands due to atomic and molecular transitions (mostly those of $O_2$ and $H_2O$), and the transmission bands in between those absorption bands. Distances between each

pair of the resulting 200 dimensional vectors could be computed to cluster or group together the most similar spectra, but rather than computing the whole matrix it is much more efficient to compute the distance between each spectrum and a tentative set of cluster centers. In this widely used method known as $k$-means, the position of each center is iteratively refined by computing the average of all spectra closest to the center. The performance of several variations of the $k$-means method is compared in Supplementary Figure 1 and Supplementary Table 1. We have empirically found that the accuracy of the method improves substantially if each spectra is normalized (divided by its vector length) before the $k$-means clustering, and converted back to the original values after a cluster has been assigned to each input spectrum. This is necessary to avoid the total integrated irradiance having too much weight during the spectrum classification, as the multijunction efficiency is most sensitive to the spectral distribution due to current matching constraints. We have also found that the number of needed proxy spectra can be reduced substantially by merging the smallest clusters with their nearest neighboring clusters, as $k$-means tends to produce a fraction of small clusters having a small impact on the yearly energy yield. In this final step, additional merging with the neighboring clusters reduces the number of obtained proxy spectra by a factor of approximately 2/3. Other clustering methods are evaluated in Supplementary Note 2 and Supplementary Figure 2.

The criteria chosen for classifying the spectra in the binning method has been the ratio of the photon flux at wavelengths longer than 650 nm to the photon flux at wavelengths shorter than 650 nm (EPR650)[2].

**Detailed balance model and radiative coupling**. As in the work by McMahon et al., the external radiative efficiency (ERE) is assumed to be 0.01 for junctions without a back mirror, the $\beta$ radiative coupling parameter is assumed to be given by $n^2 = 11$, and the recombination current is given by an ideal diode equation. We also use the same common approximation for the detailed balance radiative saturation current under the assumption of high EQE and the thermal energy being negligible in comparison with the band gap[7]. The ERE value of 0.01 used in this work is not representative of the particular case of GaAs and other high-quality III−V semiconductor single junction devices that typically operate with negligible nonradiative recombination[26]. We have therefore exceptionally used a value of ERE = 0.2 for single junctions other than silicon in Table 1. The voltage is obtained as a function of current as:

$$V = \sum_i \frac{kT}{q} \log\left(\frac{I_i - I}{I_0} + 1\right) - RI, \tag{1}$$

$$I_0 = (1 + \beta) I_{db} / ERE, \tag{2}$$

$$I_{db} = \frac{2\pi q (kT)^3}{h^3 c^2} \left(\left(\frac{E_g}{kT} + 1\right)^2 + 1\right) e^{-E_g/kT}, \tag{3}$$

where $I_i$ are the photocurrents of each junction, $R$ is the series resistance, $q$, $h$, and $c$ are fundamental constants, $kT$ is the thermal energy, and $E_g$ is the band gap. When comparing with previous calculations based on the assumption of purely radiative recombination, using realistic values of the ERE favors higher energy band gaps. Recent developments in rear heterojunction designs have led to experimental ERE values near the internal radiative limit[26]. We include an ideal back mirror for each junction at the bottom of a series connected stack, increasing the ERE for such junction by a factor $(1 + n^2) = 12$. Radiative coupling is included self-consistently. The radiative coupling depends on the maximum power point, and the maximum power point depends on the radiative coupling. The calculation is repeated until there is a negligible shift in the maximum power point.

As demonstrated by Geisz et al., shunt resistance effects are negligible in high-quality devices[13]. The effects of reverse bias breakdown are also negligible near the maximum power point under nearly current matched conditions, which are the cases of interest in this work. Series resistance cannot be overlooked, as it imposes a higher penalty on devices based on low energy band gaps. As we are interested in the upper practical limits of efficiency, we have assumed an optimistic series resistance $R = 5$ mΩ cm$^2$ for concentrator devices and 0.4 Ω cm$^2$ for one sun devices. For comparison, in the concentrator device efficiency projections made by Geisz et al., the range of values considered was 5−20 mΩ cm$^2$[15]. For our purposes, the photoluminescence can be safely assumed to be negligible in comparison with the electroluminescence near the maximum power point.

To verify our extended detailed balance model we calculate the efficiency of the current world record solar cell, which has a slightly suboptimal band gap combination due to the choice of non-alloyed GaAs for the second subcell[27]. Our calculations predict 46.58% efficiency at 508 suns under standard conditions (all other parameters being those previously mentioned), while the experimentally reported value is 46.0%. The small difference is probably due to the fact that we do not include tunnel junction losses in our calculations. Including the effects of spectral and thermal variations yields a much lower yearly averaged efficiency of 42.6%. The loss due to spectral variations is 2.18%, and the loss due to higher cell operating temperatures is 1.69%.

**External quantum efficiency model**. Most of the previous works on photovoltaic efficiency limits assume a 100% EQE at all wavelengths, but this assumption is not realistic. High band gap solar cells currently have quantum efficiencies below 80% at 400 nm, below 50% at 350 nm, and below 30% at 300 nm[17]. In contrast, the best Si solar cells have quantum efficiencies above 80% up to 320 nm due to much lower near surface light absorption (a consequence of the indirect band gap) and the excellent surface passivation attainable in this material. An ideal 100% EQE at all energies above the band gap has been assumed for silicon-based single junctions, but in all other cases, to include realistic photocurrent losses at both ends of the spectra, the following EQE model has been used:

$$EQE = \alpha e^{-\left(\frac{E - \mu}{\sigma}\right)^4} + \beta, \tag{4}$$

where $E$ is the photon energy and the parameters $\alpha = 74.53\%$, $\beta = 19.92\%$, $\mu = 1.782$ eV, and $\sigma = 1.384$ eV have been determined by fitting (in the $300-1700$ nm range) to the total EQE of the current record efficiency multijunction[27]. Each spectra is multiplied by the total EQE before integration to determine the photon flux available at each subcell. Undoubtedly, new and improved surface passivation techniques will be developed in the future, as well as new window layer materials and ARCs[28], and highly transparent top contacts[29]. But the present model can still be viewed as a moderately optimistic upper limit due to other economical and technical constraints that we do not consider explicitly. The technologies and materials that can be used in flat plate solar panels operating at 1 sun are very much constrained by cost considerations. On the other hand, concentrators introduce optical losses, mostly at the highest and lowest energies, that are optimistically modeled here with a spectrally flat transmittance of 90%.

Each subcell is assumed to absorb a fraction $(1−T)$ of the photons with energies above its band gap and none of the photons with energies lower than its band gap. The transmission factor $T$ has been set to 2%, corresponding approximately to the transmission of a 3-μm-thick GaAs subcell (the exact value ranges from 2 to 3% depending on the integration range).

**Subcell interconnection**. The efficiencies and energy yields reported here slightly overestimate the benefit of increasing the number of junctions, as tunnel junction losses are not included in our model. State-of-the-art tunnel junctions lead to electrical losses under high concentration representing less than a 0.1% absolute efficiency loss for each junction[30]. Optical losses caused by absorption in tunnel junctions are of a similar magnitude, while reflection losses can be mitigated with a proper optical design, but still represent an important loss mechanism associated with tunnel junctions[31]. The losses associated with intermediate terminals have not been explicitly included as they are expected to be small compared to top contact losses due to the reduced photon flux, reduced spectral bandwidth, longer wavelengths, and much reduced current (if using three terminals rather than four).

**Temperature**. Operation temperature variations have an effect on reverse saturation currents, but the lower ambient temperatures during sunrise also exacerbate the current mismatch due to the blue-shift of the band gaps while the solar spectrum is red-shifted[32]. This later effect has been included using the Varshni parameters for GaAs, as these are, to a first approximation, valid for most photovoltaic materials[33]. Operation temperature can be assumed to be a linear function of irradiance, as in ref. [2]. Following the finite element calculations by Marta Victoria et al.[32], a 70 °C operation temperature has been assumed as representative of small (1 mm$^2$) cells with excellent passive cooling (directly attached to a copper plate) under high concentrated irradiance (1000 suns)[32]. This is a best case scenario with current technology, but choosing such a low operation temperature might be justified as reliability studies have shown that good heat extraction can prolong the life of concentrator solar cells by decades[34]. A minimum temperature at zero irradiance of 15 °C is assumed.

## Data availability
All the code and data used in our calculations, with instructions to reproduce the results presented here, are available as open source. https://doi.org/10.5281/zenodo.1466974, https://github.com/Ripalda/Tandems).

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

## Acknowledgements

We thank Dr. Jose M. Llorens (IMN-CSIC) for advice and assistance. Funding was provided by MINECO (TEC2015-64189-C3-2-R) and Comunidad de Madrid (S2013/MAE-2780). I.G. is funded by Ministerio de Economía y Competitividad (RYC-2014-15621).

## Author contributions

J.M.R. wrote the initial version of the manuscript and python code. J.B. contributed to the methodology (machine learning, detailed balance) and improved the python code. I. G. contributed to the methodology (binning, temperature effects) and all authors jointly contributed to the manuscript.

## Additional information

**Competing interests:** The authors declare no competing interests.

