## [Peer Review File · Nature Communications]

Reviewers' comments:

Reviewer #1 (Remarks to the Author):

The paper claims that applying a machine learning algorithm called K-means clustering will improve the yearly prediction accuracy of averaged photovoltaic efficiency. The method is validated based on a calculated dataset with a number of random 6 junction band gap combinations, and compared with a commonly used binning method. In addition, the proposed machine learning method is integrated multijunctions with silicon bifacial single junctions so that the energy can generated more than 1 MWh/m². The paper is well written and seems like exceed the state-of-the-art. There is only one minor question that authors should address is the rational of choosing K-means clustering method. There are many other clustering methods such as support vector machines etc. Is K-means method randomly chosen or other methods have been tested and K-means is the best among the rest.

Reviewer #2 (Remarks to the Author):

The manuscript starts by describing the existing gaps and limits of the current PV technology and the dependency of their efficiency from spectral sensitivity effects. The innovative element the authors introduce here is the use of Machine Learning (ML) techniques for deriving a minimal representative set of yearly spectra for a more accurate and precise estimate of efficiency (or yearly energy yield) of photovoltaic devices. This approach classifies and clusters yearly spectra into few categories, claiming a reduction of about three orders of magnitude with respect to the full yearly set of spectra. The authors show how this new classification allows reaching a more accurate estimate of the yearly energy efficiency than a common binning-averaging method. The outcome (proxy set of spectra) of this ML approach is then used to perform solar cell band gaps optimization by computing and maximizing energy yield instead of the commonly used efficiency under standard conditions. With this approach the authors show that the number of junctions is not the most important parameter towards maximizing the energy yield of a solar device.

The topic of the paper is of interest to the community and the proposed method can be used to define new standards for an accurate determination of the yearly energy yield of a PV device. The title of the manuscript focuses the attention on ML, but from the manuscript it looks like this technique has been used only for a preparatory part, i.e. reducing the dimensionality of the problem, and not for the direct calculation of the energy yield itself (i.e. authors are not using regression techniques or similar). In this respect, the title can be maybe revised, opting for a choice which is less ML-oriented.

The manuscript needs some revisions on the way it is written (authors are encouraged to run a spell checker) and on the way the topics are organized. I offer a few suggestions in the attached pdf document, but the text needs major improvements in my view. Furthermore, in addition to the description of the method, one or two intrinsic validation plots about the used machine learning classification technique would be useful to add. For example, could the authors plot a sort of misclassification error as a function of neighbors k , to justify your final choice of the number of clusters? This plot could well fit under the paragraph "Machine learning", for example. Or is it Figure 2 already supposed to show this misclassification in terms of efficiency overestimate?

Finally, the results seem to be fully reproducible, as the authors also provide the link to the python code developed for obtaining the results described in the manuscript.

For more detailed comments please see the attached document.

Madrid, September 19th, 2018

Reviewer #1.: "Is K-means method randomly chosen or other methods have been tested and K-means is the best among the rest?"

We thank the reviewer from bringing up this important issue. The k-means method was chosen after extensive testing of a wide range of machine learning methods. The details of our testing can be found in a new supplementary material file attached to this resubmission.

Reviewer #2.: "The title of the manuscript focuses the attention on ML, but from the manuscript it looks like this technique has been used only for a preparatory part, i.e. reducing the dimensionality of the problem, and not for the direct calculation of the energy yield itself (i.e. authors are not using regression techniques or similar). In this respect, the title can be maybe revised, opting for a choice which is less ML-oriented."

We agree with the referee that the previous version of the title could cause confusion as to how exactly we were using machine learning. To make it clear that machine learning has been used only to describe spectral variations, we have changed the title of the manuscript to "Solar cell designs maximizing annual energy production based on machine learning of spectral variations". Hopefully this is a better description of our work.

R.#2: "For example, could the authors plot a sort of misclassification error as a function of neighbors k , to justify your final choice of the number

of clusters? This plot could well fit under the paragraph 'Machine learning?', for example. Or is it Figure 2 already supposed to show this misclassification in terms of efficiency overestimate?"

The new figure suggested by reviewer #2 has been included as Fig. 2. The figure formerly included as Fig. 2 (now Fig. 3) does indeed give some information on the misclassification error in terms of the efficiency overestimate, but we agree with reviewer #2 on the need to plot a metric more directly related to the misclassification error (such as the RMS), as the efficiency overestimate also depends on other factors such as cluster size homogeneity.

R.#2: "Are we sure that the convergence is at 15 (number of clusters for spectra adopted for the study)? What happen for # spectra above 20? Could you extend a bit more the graph (let's say up to 30)? And what drives your choice of 15?"

As suggested by reviewer #2, we have extended our study up to 30 clusters. The new results show that the quality of the clustering results keeps improving as the number of proxy spectra (clusters) is increased beyond 20, which was the higher number of clusters in the previous version of the manuscript, while the binning method remains stalled. Using a single average spectrum, the efficiency overestimate is 3% in the worst case. Our (neccessarily arbitrary) convergence target was for the efficiency overestimate to be reduced by an order of magnitude, from 3% to 0.3%, and this criteria was met with 15 clusters in the most difficult case (6-junctions in series).

R.#2: "Why the efficiency here [average eff. for diffuse irradiance] is higher than the standard efficiency?"

We thank the reviewer for pointing this out. As there is no defined standard for diffuse spectral irradiance, we no longer report a standard efficiency value for diffuse irradiance. The previously reported standard efficiency value for diffuse irradiance was based on the difference spectrum between the global and direct standard spectra, but this was a mistake, as

the geometry defined in both standards is different (tilted plane vs. normal incidence). The yearly averaged efficiencies are calculated by subtracting the direct normal spectral irradiance from the global spectral irradiance on a sun tracking surface; in this case the geometry is the same.

R.#2: "Do you also include the Machine Learning code? I couldn't find any link on GitHub regarding this part."

Our machine learning code is in the open source file `/src/tandems.py` at <https://github.com/Ripalda/Tandems>. The exact location can be found by searching for "machine learning" from within the `tandems.py` file. The `tandems.py` code imports the open source sklearn machine learning library available at <http://scikit-learn.org/>. These details are now part of the manuscript.

To avoid redundancy, some reviewer's comments and suggestions are not directly addressed here, but are reflected in the corresponding manuscript changes (highlighted for clarity).

Sincerely,

Jose M. Ripalda, Jeronimo Buencuerpo, Ivan Garcia

REVIEWERS' COMMENTS:

Reviewer #1 (Remarks to the Author):

The authors have successfully addressed all my questions.

Reviewer #2 (Remarks to the Author):

Dear authors of NCOMMS-18-16540,

I would like to thank you for addressing the comments and for clarifying the questions raised during the first review.

My last suggestions, after reading the new version of the manuscript:

- Would it be acceptable to modify the title to: "Solar cell designs by maximizing annual energy production based on machine learning clustering of spectral variations"?

- In the Abstract: "Here we demonstrate machine learning techniques for reducing..."  "Here we demonstrate how machine learning techniques can reduce..."

- Github code: I suggest creating a persistent DOI for your GitHub code through an open-science platform like Zenodo (<https://zenodo.org/account/settings/github/>). Afterward, you can directly refer and include your DOI in the manuscript references and also it will make your code citeable for other authors in the future.

Other than these little remarks, I believe the manuscript is ready to go through the next publication steps.

Congratulations on this nice work.